# Thermokarst landscape exhibits large nitrous oxide emissions in Alaska's coastal polygonal tundra
Josh Hashemi [1,2,3] ✉, David A. Lipson [1], Kyle A. Arndt [4], Scott J. Davidson[5], Aram Kalhori[6], Kyle Lunneberg[1], Lona van Delden[3], Walter C. Oechel [1,7,9] & Donatella Zona [1,8,9]

Global atmospheric concentrations of nitrous oxide have been increasing over previous decades with emerging research suggesting the Arctic as a notable contributor. Thermokarst processes, increasing temperature, and changes in drainage can cause degradation of polygonal tundra landscape features resulting in elevated, well-drained, unvegetated soil surfaces that exhibit large nitrous oxide emissions. Here, we outline the magnitude and some of the dominant factors controlling variability in emissions for these thermokarst landscape features in the North Slope of Alaska. We measured strong nitrous oxide emissions during the growing season from unvegetated high centered polygons (median (mean) = 104.7 (187.7) $\mu g$ $N_2O$-N $m^{-2}$ $h^{-1}$), substantially higher than mean rates associated with Arctic tundra wetlands and of similar magnitude to unvegetated hotspots in peat plateaus and palsa mires. In the absence of vegetation, isotopic enrichment of $^{15}N$ in these thermokarst features indicates a greater influence of microbial processes, (denitrification and nitrification) from barren soil. Findings reveal that the thermokarst features discussed here (~1.5% of the study area) are likely a notable source of nitrous oxide emissions, as inferred from chamber-based estimates. Growing season emissions, estimated at 16 (28) mg $N_2O$-N $ha^{-1}$ $h^{-1}$, may be large enough to affect landscape-level greenhouse gas budgets.

Greenhouse gas (GHG) dynamics in permafrost ecosystems have been shifting due to increasing temperatures, active layer thickness, and hydrology with positive feedbacks on warming[1]. As permafrost soils make up one of the largest terrestrial reservoirs of carbon (C) and nitrogen (N)[2–4], accumulated over millennia due to cold, water-saturated soils with slow decomposition[5], increased attention has been given to GHG dynamics in permafrost regions over recent decades[6–13]. The majority of regional GHG studies have focused on C emissions (i.e., carbon dioxide ($CO_2$) and methane ($CH_4$)), outlining effects of, among others, seasonality[9,10,14,15], landscape heterogeneity[16,17], vegetation composition[18] and vegetation density[19]. However, few studies have reported the flux dynamics of nitrous oxide ($N_2O$), an ozone depleting substance and powerful GHG with a 100-year global warming potential ($GWP_{100}$) 273 times that of $CO_2$[20].

$N_2O$ is produced from various biological and chemical processes happening simultaneously in the soil[21,22]. Production pathways of $N_2O$ are predominantly via nitrification, where $N_2O$ is a by-product in the oxidation of ammonium ($NH_4^+$) to nitrate ($NO_3^-$), and denitrification, where $N_2O$ is an intermediate in the reduction of nitrite ($NO_2^-$) and $NO_3^-$ to produce dinitrogen ($N_2$) gas[21]. These two processes are interconnected and mainly driven by temperature, oxygen availability, and substrate availability and are limited in high-latitude ecosystems with short growing seasons[21]. $N_2O$ emissions have therefore often been considered negligible in permafrost regions, because of limited mineral N availability due to cold and wet environmental conditions[21]. Warming can lead to increased decomposition, mineralization, and release of N, previously locked in organic matter rich, permafrost-dominated Arctic soils[21,23–25]. Additions of this released bioavailable N can then act as a substrate for increased $N_2O$ production.

Strong plant competition for available inorganic N can reduce the production and emission of $N_2O$ in vegetated areas[26]. Plants ultimately absorb most of the bioavailable N due to the greater N demand by plants compared to the supply[27,28]. High microorganism turnover (3–5 days) results in a redistribution of soil N while plants slowly accumulate large

[1]Biology Department, San Diego State University, San Diego, CA, USA. [2]Department of Land, Air and Water Resources, University of California Davis, Davis, CA, USA. [3]Alfred Wegener Institute Helmholtz Centre for Polar and Marine Research, Potsdam, Germany. [4]Woodwell Climate Research Center, Falmouth, MA, USA. [5]University of Plymouth School of Geography, earth and environmental sciences, Plymouth, UK. [6]GFZ German Research Centre for Geosciences, Potsdam, Germany. [7]Department of Geography, University of Exeter, Exeter, UK. [8]Department of Animal and Plant Sciences, University of Sheffield, Sheffield, UK. [9]These authors contributed equally: Walter C. Oechel, Donatella Zona. ✉e-mail: joshua.hashemi@awi.de

portions of available N due to a lower turnover (1–3 months)[28]. However, unvegetated areas, common in Arctic regions[29–32], remove competition for N by vascular plants and can result in higher rates of $N_2O$ production due to increased inorganic N availability for nitrification and denitrification processes[33,34].

Emissions of $N_2O$ can be difficult to capture due to their high spatial and temporal variability on the landscape scale as well as on the micro-scale. This is due, in part, to $N_2O$ production happening during both aerobic and anaerobic soil conditions, which can simultaneously occur on the microscale in the highly complex soil matrix[35]. With increasing water filled pore space (WFPS), $N_2O$ production can shift from nitrification to incomplete denitrification, provided there is adequate $NO_3^-$ available, and eventually ends in $N_2$ release when the soil is completely water saturated, limiting oxygen availability[22]. These dynamic processes contribute to the difficulty in upscaling $N_2O$ budgets, particularly in remote Arctic regions, as data collection campaigns may be sparse. Despite this, some research suggests that $N_2O$ emissions from permafrost ecosystems may have a significant and growing impact on the global $N_2O$ budget, contributing 0.14–1.27 Tg $N_2O$-N per year (7% of global budget)[21]. Increasing soil temperatures and associated hydrological changes may facilitate conditions favorable for increased N cycling[36]. Given this potential positive feedback on warming, a better understanding of the response of $N_2O$ dynamics to warming and associated environmental changes in permafrost regions is needed.

Despite the number of studies on GHG fluxes, few in-situ $N_2O$ measurements have been published from the North Slope of Alaska[21]. The North Slope of Alaska is comprised of a patchwork of landscape features that includes lakes, ponds, drained lake basins, drained upland tundra and polygonal tundra with high levels of organic C and N[37–39]. Polygonal tundra are characterized by surface relief created by the common development and growth of ice wedges. Over time, these ice wedges lift areas of the soil, creating ridges and forming complex wetlands with distinct polygonal patterns that vary in position of the water table[40].

Landscape heterogeneity in this region is due, in part, to freeze-thaw dynamics[40]. In particular, polygonal tundra, extending over an estimated 65% of the Arctic Coastal Plain[37] and covering 3% of Arctic landmass (~250,000 $km^2$)[41], can result in significant variability in vegetation composition[42,43], hydrology[38,40,44,45], GHG dynamics[38,46,47], and a wide range of GHG budget estimates[10,17,48,49]. Complex interactions of hydrology, ice wedge dynamics, and freeze-thaw cycles result in high-centered polygons, i.e., soil mounds that protrude above the water table[40,50]. Cryoturbation, thaw processes, thermal erosion, and changes in hydrology can destabilize and shift overlying soil structures[51,52] disrupting the rooting structures of vascular plants, and causing high-centered polygons to degrade[53]. This can result in thermokarst-affected high-centered polygon features with exposed and unvegetated soil (hereafter referred to as "thermokarst polygons") that can increase the rate of mineralization of N and affect plant-microbe competition for inorganic N[21,54]. Notably, the complex feature mosaic of the North Slope of Alaska landscape has been identified to have a high $N_2O$ potential from airborne eddy covariance screenings[55] with the source feature remaining unknown.

The aim of this study was to identify the role of progressive thermokarst development and thermal erosion on $N_2O$ emissions with the expectation that areas with little or no vegetation within polygonal tundra of the North Slope of Alaska exhibit similarly high levels of $N_2O$ emission, comparable to the previously identified peat circle (Russia)[29] and palsa (Finland)[30] hotspots. Further, we address whether these polygon features have a larger climate forcing potential than previously assumed, when accounting for the much larger warming power of $N_2O$ compared to $CO_2$[20]. Our estimates of chamber based $N_2O$ emissions, in conjunction with previous airborne eddy covariance measurements[55] indicate that these thermokarst polygons and resulting bare spots may be intensive enough to affect the GHG budget on the landscape scale of the North Slope of Alaska.

## Results and discussion
### In-situ GHG flux measurements

GHG fluxes were estimated using the static chamber technique on the Barrow Environmental Observatory (BEO), a polygonal tundra south of Utqiaġvik, Alaska (Fig. 1a). We report fluxes of both $N_2O$ and $CO_2$ (Net Ecosystem Exchange (NEE)) to show the combined climate forcing potential of two of the main GHGs in these high latitude systems. Measurements were taken at thermokarst polygon surfaces (Fig. 1b; Supplementary Fig. 1) in unvegetated (Fig. 1c), and nearby vegetated areas (Fig. 1d) during the growing season (July). Vegetated and unvegetated features experienced a similar range of water table, soil water content, soil temperature, and thaw depth (Supplementary Fig. 2). Stable isotope and carbon to nitrogen (C:N) ratios were measured to support the interpretation of GHG flux dynamics by site and soil depth relating to the influence of the presence of vegetation cover.

Unvegetated areas (number of measurement locations = 20) on thermokarst polygons show significantly higher ($p < 0.001$) emissions of $N_2O$ (median (mean ± standard error) = 104.7 (187.7 ± 17.4) µg $N_2O$-N $m^{-2} h^{-1}$) in comparison with vegetated areas (number of measurement locations = 10; 13.5 (34.2 ± 12.1) µg $N_2O$-N $m^{-2} h^{-1}$) (Fig. 2a). The emissions from unvegetated areas reported here are more than two orders of magnitude higher than the median (mean) rate associated with permafrost wetlands (0.8 (5.2) µg $N_2O$-N $m^{-2} h^{-1}$) and substantially higher than emissions measured from Arctic peatlands and upland tundra (2.5 (24.8) & 1.4 (8.8) µg $N_2O$ -N $m^{-2} h^{-1}$, respectively)[21]. Emissions are also higher than those reported from unvegetated areas in permafrost regions in general (18 (42) µg $N_2O$-N $m^{-2} h^{-1}$)[21], and close to mean emissions found from previously identified Arctic $N_2O$ hotspots from unvegetated peat circles (~230 µg $N_2O$ -N $m^{-2} h^{-1}$)[29] and palsa mires (~270 µg N $N_2O$ -N $m^{-2} h^{-1}$)[30]. $N_2O$ emission rates from unvegetated surfaces of thermokarst polygons are comparable to mean tropical organic soils (up to 125 µg $N_2O$-N $m^{-2} h^{-1}$)[56], highlighting the importance of permafrost regions and Arctic tundra in the global $N_2O$ cycle. Analysis of UAV imagery across the study area reveals that approximately ~1.5% of the land surface consists of these unvegetated features (Supplementary Fig. 3). Based on this data, area-adjusted estimates suggest midday $N_2O$-N emissions are around 16 (28) mg $ha^{-1} h^{-1}$ during the growing season. Although emissions from barren regions reported here do not account for the full diurnal cycle (measurements took place between 9:00 and 16:00), clear partial diurnal trends are observed (Supplementary Fig. 4) illustrating the importance of diurnal fluctuations for $N_2O$ emissions estimates.

Emission rates of $N_2O$ at vegetated areas on thermokarst polygons are also slightly higher in comparison to the above estimates for permafrost wetlands. High-centered polygons are better drained than surrounding lower areas that are commonly inundated. Therefore, these features are better aerated and have higher oxygen availability than the surrounding permafrost wetlands. The greater oxygen concentrations and higher decomposition would be expected to result in greater N availability and $N_2O$ production via nitrification. Cryoturbation and freeze-thaw cycles in high-centered polygons may also mix N from deeper soil layers near or at the permafrost table. This can mobilize existing pockets of $N_2O$ and inorganic nitrogen within the permafrost, bringing them closer to the active layer for potential uptake thereby increasing mineral N availability for $N_2O$ production[23,24,57,58]. In addition, vegetation communities on these structures contain moss and lichen communities that are associated with biological nitrogen fixation. This can increase the soil inorganic N pool[59,60] and possibly - through increased mineral N availability - $N_2O$ production, relative to inundated areas[22].

Measurements of NEE reveal significantly larger $CO_2$ emissions from unvegetated surfaces ($p < 0.001$, DF = 28), showing these areas to be a source median (mean ± standard error) = (36.2 (38.0 ± 1.88) mg C-$CO_2$ $m^{-2} h^{-1}$) compared to a weak sink at adjacent vegetated areas ($-7.7$ ($-6.6 ± 0.08$) mg C-$CO_2$ $m^{-2} h^{-1}$) during the daytime (Fig. 2b). The difference is likely primarily due to the absence of $CO_2$ uptake through photosynthesis at the unvegetated surfaces, resulting solely in ecosystem respiration (ER). Fluxes of $CO_2$ from vegetated and unvegetated areas are similar to previous

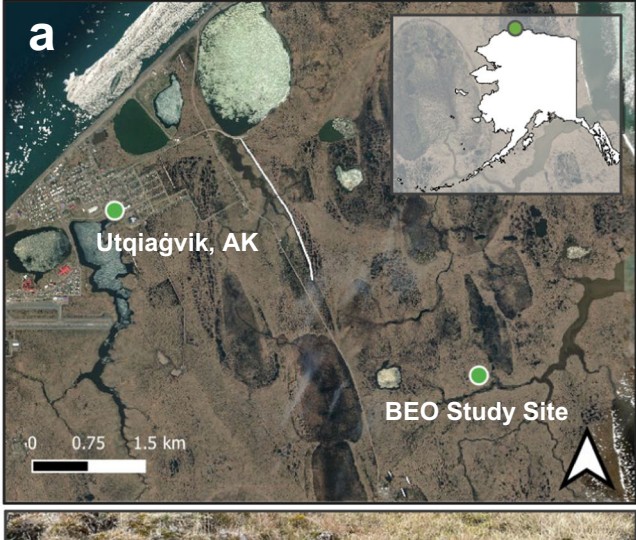

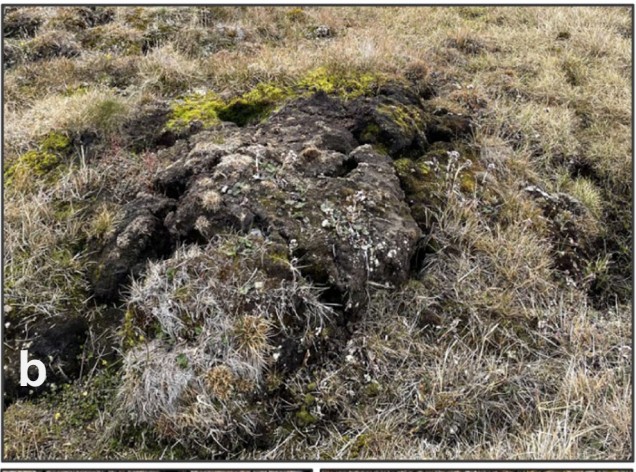

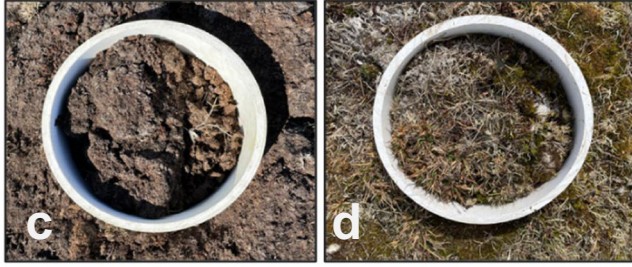

**Fig. 1 | Overview of the site location and studied landscape features. a** Study area near Utqiaġvik, AK and (**b**) eroding high center polygonal landscape feature with photos of collars with (**c**) unvegetated and (**d**) vegetated surface. Map (**a**) source credits: Esri, Maxar, GeoEye, Earthstar Geographics, CNES/Airbus DS, USDA, USGS, AeroGRID, IGN and the GIS User Community.

estimates of NEE[61] and ER[62] respectively. When estimating the $GWP_{100}$ of thermokarst polygons, the combined effect of $CO_2$ and $N_2O$ nearly doubled the climate forcing potential (64.58 (89.07 ± 6.6) mg $CO_2$eq $m^{-2}$ $h^{-1}$) when compared to that of $CO_2$ alone. This comparison only reflects midday conditions and does not take the diurnal variability of both $CO_2$ and $N_2O$ into account. It is possible that the relative effect of $CO_2$ emissions would be higher if diurnal variability were considered, though the variability in $N_2O$ diurnals is unknown.

### Variability in and controls on $N_2O$ emission strength

Linear mixed effects model output indicates that higher $N_2O$ emissions at unvegetated areas are associated with lower WFPS and tend toward higher temperatures (Fig. 3, Supplementary Table 1). No relationships between ancillary data and $N_2O$ fluxes were found to be significant in vegetated areas.

The difference in patterns of emissions among vegetated areas and unvegetated areas is likely due, in part, to reduced competition for inorganic N by vascular plants, relatively warm soil temperatures and increased oxygen availability due to greater soil aeration. Bulk density was significantly lower in unvegetated soil (0.073 ± 0.004 g $cm^{-3}$) than in vegetated areas (0.119 ± 0.006 g $cm^{-3}$), possibly allowing for increased oxygen penetration into the soil column and thus, conditions more favorable for nitrification, at least in the upper 15 cm where reported soil water content was measured. $N_2O$ emissions peaked at around 20% WFPS in the top 15 cm (Fig. 3). While this may provide support for nitrification-based $N_2O$ production, denitrification has been identified as a dominant $N_2O$ emission pathway for tundra regions[34] and barren mineral polygon tundra[63], and thus likely occurs at deeper soil layers (>15 cm) that have increased soil water content and limited oxygen availability. The significant interaction of WFPS and soil temperature ($p = 0.02$) in the highest performing model (pseudo $R^2$ (fixed effects) = 0.34) may give some indication of this as well, as higher surface temperatures with lower WFPS could be correlated with warmer conditions in deeper soil promoting enhanced denitrification. Though there is an obvious influence of temperature over the microbial processes governing $N_2O$ emissions, in permafrost regions, $N_2O$ emissions have been found to be dominantly controlled by substrate availability and conditions associated with oxygen availability, such as soil moisture and soil pore size[64]. WFPS is tightly related to soil redox potential and oxygen availability, as soil diffusivity increases with lower bulk density and lower soil water content[22]. Thaw depth showed no correlation (Supplementary Fig. 5) and decreased multivariate model performance (Supplementary Table 1). Stronger correlations of $N_2O$ emissions with thaw depth would be expected with permafrost thaw due to the introduction of new organic matter rather than seasonally thawing active layer[65].

### Carbon and Nitrogen composition of soil environment

Soil samples from areas with no vegetation were significantly higher in both δ $^{15}$N (30 ± 2.34‰) and δ $^{13}$C (−4.74 ± 3.2‰) content than in vegetated soils (δ $^{15}$N: 14.58 ± 3.3‰; δ $^{13}$C: −21.32 ± 2.8‰) (Fig. 4a, b). The elevated δ $^{13}$C signature of unvegetated soils may be due to (1) the influence of microbial products derived from older labile plant compounds, combined with the preferential loss of lighter carbon over time and a lack of seasonal inputs, (2) localized carbonate accumulation or (3) a combination of these two processes. Isotopic enrichment of $^{15}$N in unvegetated soil areas indicates a larger loss of gaseous N species via microbial processes such as nitrification and denitrification, as plant uptake does not occur[34,66]. Ammonium volatilization is likely limited due to the acidic soils in this region[67], however may occur to some extent if unvegetated areas exhibit localized increased alkalinity. Both nitrification and denitrification are highly sensitive to changes in oxygen availability and due to the variable nature of hydrology in polygonal tundra[40,68], microsite variability in moisture content may support high rates of $N_2O$ production through both of these pathways simultaneously. It is possible that emissions of $N_2O$ from thermokarst polygons were primarily from nitrification due to the strong relationship with properties governing oxygen availability. However, at deeper, more saturated soil levels closer to the permafrost table, oxygen availability is likely more limited and could allow for denitrification or nitrifier denitrification to substantially contribute to surface flux[34]. Particularly noteworthy are transition zones, where abrupt changes in oxygen availability occur. These transition zones may create conditions suitable for simultaneous denitrification and nitrification, and contribute to elevated $N_2O$ emissions from these specific microenvironments. Deeper areas, nearer to the permafrost table could also contribute the overall $N_2O$ emissions through available $NO_3^-$ and/or $NO_2^-$ release directly from the permafrost.

Data from δ $^{15}$N and δ $^{13}$C in unvegetated areas showed no significant relationship with depth in the top 15 cm of the soil column (Supplementary Fig. 6) though the deepest soil layer in vegetated areas had elevated δ $^{13}$C, possibly indicated some level of freeze thaw mixing with nearby unvegetated soils. Although mean values for δ$^{15}$N (~33‰) were higher at deeper areas compared to shallower depths (~29‰), these differences lacked significance

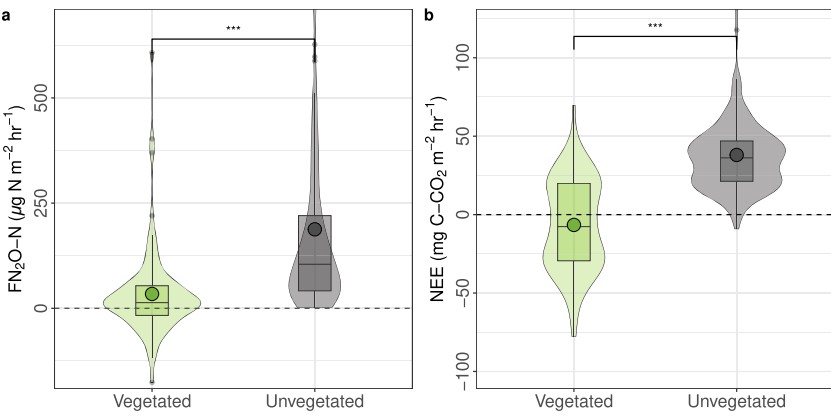

**Fig. 2 | Nitrous oxide and carbon dioxide fluxes at vegetated and unvegetated areas.** Comparison of (**a**) FN$_2$O-N ($\mu$g N m$^{-2}$ h$^{-1}$) and (**b**) FC-CO$_2$ (NEE) (mg C m$^{-2}$ h$^{-1}$) at areas with vegetated and unvegetated soil surfaces. Circles represent means and violin plots indicate the distribution of data. $n = 182$ at unvegetated sites $n = 80$ at vegetated sites. Asterisks indicate significance value: *** = $p < 0.001$ (ANOVA). Some positive outliers in unvegetated areas were not included in figures for better graphical representation.

due to the broad ranges observed at shallower depths. This may suggest a correlation with soil layers deeper than those measured here, potentially supporting increased denitrification in deeper soil layers. Although previous research has demonstrated a relatively uniform distribution of sequences corresponding to denitrification respiratory pathway genes with soil depth in this locale[69], nitrate profile analysis from polygon rims show low nitrate at the deepest soil layers, as nitrate is rapidly reduced in these suboxic environments[70]. The high variability of these data likely reflects that the pathways of N loss leading to isotopic enrichment are episodic and therefore highly variable.

Significantly lower C:N ratios ($p = 0.03$) were found in unvegetated soils (14.82 ± 0.52) than in vegetated (17.63 ± 1.1) (Fig. 4c). This potentially provides support of greater N bioavailability for N$_2$O production in unvegetated soils[71]. Differences in C:N ratio were driven by a higher N content in unvegetated areas (17.6 ± 0.7 mg N g$^{-1}$) than in vegetated soils (10.3 ± 1.4 mg N g$^{-1}$) (Supplementary Fig. 7). Total C and N content per volume was not significantly different when standardizing with mean bulk density measurements (Supplementary Fig. 7). In vegetated soil areas, C:N ratios generally decreased with depth in the top 15 cm of the soil column (Supplementary Fig. 8), likely related to plant N uptake occurring at increased rates closer to the surface where plant root tissue is more abundant[26,27]. Mean and median total C and N were both higher at deeper soil depth in vegetated areas, albeit not significantly (Supplementary Fig. 9).

## Implications of N$_2$O emissions on the landscape scale

Data presented here have important implications for regional estimations of future N$_2$O emissions due to substantial hydrological, thermal, active layer, and land surface changes expected in high latitude ecosystems in coming decades. Arctic wetlands maintain water tables near or above the soil surface for most or all of the year, due to limited drainage created by the permafrost barrier, facilitating anaerobic conditions in the soil column[67]. However, as regional warming continues, permafrost degradation could cause increased active layer depths, lateral movement of water and drainage of polygonal tundra[40,72]. These processes heighten the likelihood of barren soil exposure due to thermokarst and thermal erosion, evident in a sixty-fold surge in the development and expansion of retrogressive thaw slump thermokarst features in recent decades[73]. This expansion, coupled with surface disturbances, amplifies the potential for Arctic wetlands to emerge as globally significant sources of N$_2$O emissions. Additional N$_2$O flux measurements across a wide range of sites in heterogeneous tundra environments are needed to understand and document the variability of emissions due to topography, vegetation and environmental conditions. High-emitting landscape features like thermokarst polygons may substantially increase with likely changes in temperature, hydrology, and active layer depth[40].

Previous assumptions of negligible N$_2$O emissions rates from Arctic environments are increasingly challenged based on low, but evident circumarctic emissions around 1.25 $\mu$g N$_2$O-N m$^{-2}$ h$^{-1}$ [21] with an increasing body of evidence of high emission features[29,30,74] emitting up to >260 $\mu$g

N$_2$O-N m$^{-2}$ h$^{-1}$ [57]. Though low N$_2$O emissions or N$_2$O uptake driven by denitrification is often reported in high latitude wetlands[21], high landscape scale, growing season N$_2$O emissions from the North Slope of Alaska were identified using aircraft eddy covariance, showing a mean of ~99 $\mu$g N$_2$O-N m$^{-2}$ h$^{-1}$ [55]. The results presented here identify a possible contributing source of these landscape relevant N$_2$O emissions, highlighting the importance of small-scale landscape features (≤0.5 m$^2$ area). The larger distribution of thermokarst polygons across the North Slope region is unknown, primarily attributable to the challenges posed by their small size, rendering them less discernible through satellite imagery.

UAV imagery over a limited area encompassing the study region places estimates of the feature coverage at ~1.5% of the land surface, though how representative this estimate is for the North Slope region and polygonized tundra in general remains uncertain. The mean flux rate adjusted for the estimated feature coverage is 28 mg N$_2$O-N ha$^{-1}$ h$^{-1}$. This is still significantly lower than those estimated from airborne eddy covariance. The potential application of larger scale UAV imagery orthomosaics could help identify a more constrained distribution of these features and provide a means for regional upscaling of fluxes to compare emissions reported here with the previously mentioned estimates from airborne eddy covariance. Persistent disparities between these estimates may signify the existence of stronger N$_2$O emissions from thermokarst polygons not captured in the presented data. This also indicates further unaccounted-for high-emitting landscape features, such as boundary layers to water bodies or certain topography features that could result in additional N$_2$O hotspots.

While the data presented here offer insight into a novel N$_2$O source, there are several limitations and avenues for future research. In particular, data are needed on soil composition, nitrogen cycling, microbial community composition, and inorganic nitrogen content across soil depths to elucidate elevated $\delta^{13}$C, dominant N$_2$O production pathways, and identify zones of N$_2$O production. High resolution imagery over larger spatial extents are needed to improve scaling efforts as the more widely available coarser resolution imagery are unable to detect the sub-meter features discussed here. There is currently a paucity of in-situ data over longer time periods making upscaling to regional estimates very challenging. As conditions favorable for N$_2$O production and release can rapidly change[75], the emission rates observed here are only representative of midday growing season emissions.

The annual contribution to the global N$_2$O budget from these regions is still currently unknown. In particular, measurements of N$_2$O flux data from outside of the growing season are lacking. Year-round flux measurements at the landscape level, e.g., eddy covariance and automated chamber systems are needed to determine diurnal behavior and seasonal budget dynamics and to better inform model parameterizations. Arctic wetlands exhibit strong emissions of both CO$_2$ and CH$_4$ during seasonal shoulder periods, notably in the autumn zero curtain period of soil freezing[9,10,76]. As plant uptake of inorganic N should be limited outside of the growing season due to lower plant productivity and plant senescence, significant emissions of N$_2$O

may occur during this period. Likewise, data collected during the spring could reveal large budgetary contributions as the spring thaw period has been associated with peak $N_2O$ emission[32]. A budgetary understanding of regional emissions of $N_2O$ will likely increase warming potential estimates of Arctic wetlands and account for an additional warming feedback. The magnitudes of the emissions from these thermokarst polygon features highlight the importance and further need of in-situ $N_2O$ source and sink identification for future climate forcing potential from warming Arctic environments.

## Materials and methods
### Study site
This study was conducted near Utqiaġvik, Alaska in a well-developed polygonal tundra consisting of high and low-center polygons, on the Barrow

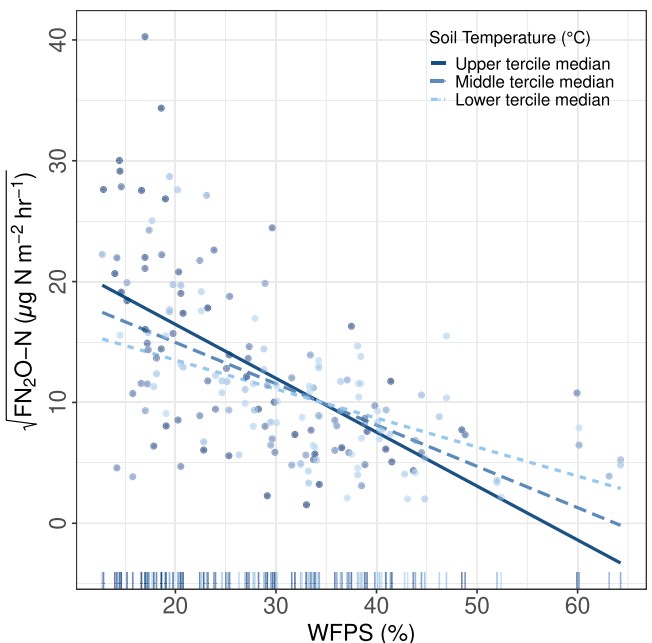

**Fig. 3 | WFPS and soil temperature controls on $N_2O$ fluxes at unvegetated areas.** The impact of the Interaction between WFPS (%) and soil temperature (°C) on $FN_2O$ ($\mu$g N m$^{-2}$ h$^{-1}$). Regression lines show a variable effect of WFPS on $FN_2O$ at the lower, middle upper tercile median. $FN_2O$ has been square root transformed to meet the normality and homoscedasticity assumptions required for analyses. The marginal rug plot above the x-axis shows the predictor relationship.

Ecological Observatory (BEO; 71 16' 51"N, 156 26' 44"W) (Fig. 1a). The BEO is on the Arctic Coastal Plain on the North Slope of Alaska and is predominantly (65%) polygonal tundra with the remainder of the landscape comprised of a combination of lakes, drained lake basins and upland tundra[37]. Soils in the BEO are in the continuous permafrost zone and are gelisols turbels (cryoturbated soils: 71–77%; orthels (mineral): 8%; organic soils: 1%) with high levels total organic C and N (18% & 0.7% respectively)[18,77]. Vegetation primarily consists of wet sedges (*Carex aquatilis*) and mosses (*Sphagnum* spp. and *Drepanocladus* spp.) in heavily inundated areas such as low center polygons and troughs, and moss/lichen (*Polytrichum* spp. & *Dicranum* spp.) dominated communities in high-center polygon and ridge areas[43]. The water table is variable depending on landscape relief and can be as high as ≥20 cm above the ground surface and as low as ≥50 cm below the ground surface. Collar locations were only in thermokarst polygons with unvegetated soil or adjacent vegetated areas also on thermokarst polygons. All thermokarst polygon features had a water table at or below the active layer depth throughout the study period. Mean maximum active layer thaw depth in these features was estimated at ~40 cm.

### GHG flux and ancillary measurements
Static chamber fluxes were measured with a Gasmet GT5000 Terra Fourier transform infrared (FTIR) GHG analyzer and a clear, cylindrical polycarbonate chamber (50 cm height and 20 cm diameter) (Supplementary Fig. 10) in a closed system at a 1 Hz sampling rate. Due to the small chamber dimensions and low temperature variability at the sampling location, no pressure vent, cooling system or fan was added based on previously established chamber designs[18,43], and relying on the pump of the GT5000 Terra to create adequate mixing within the chamber. The GT5000 Terra is capable of measuring multiple gases simultaneously by scanning the full infrared spectrum and calculating the concentrations of each gas in the sample based on its absorption with a precision of ± 3% and a minimum detectable concentration difference of 5 ppm and 7 ppb for $CO_2$ and $N_2O$, respectively[78,79]. FTIR enables the identification of unique regions with distinct peaks and characteristics within the measurement spectrum, effectively mitigating any issues related to measured gas cross-interference. Zero-point calibration was performed with N2 immediately before and after each use to ensure that any background signals or offsets in the values reported by the GT5000 Terra were minimized. Chamber collars were made of PVC (15 cm height and 20 cm diameter) and installed 3 days prior to GHG measurements at a depth of 10 cm. The chamber was ventilated prior to every measurement and placed on top of the collar ensuring an airtight connection via a rubber seal fitting the chamber to the collar (Supplementary Fig. 10). Following chamber placement, measurements were recorded over 7 min to obtain a stable increase or decrease in GHG concentration. Fluxes were calculated according to the linear slope fitting technique[80] using

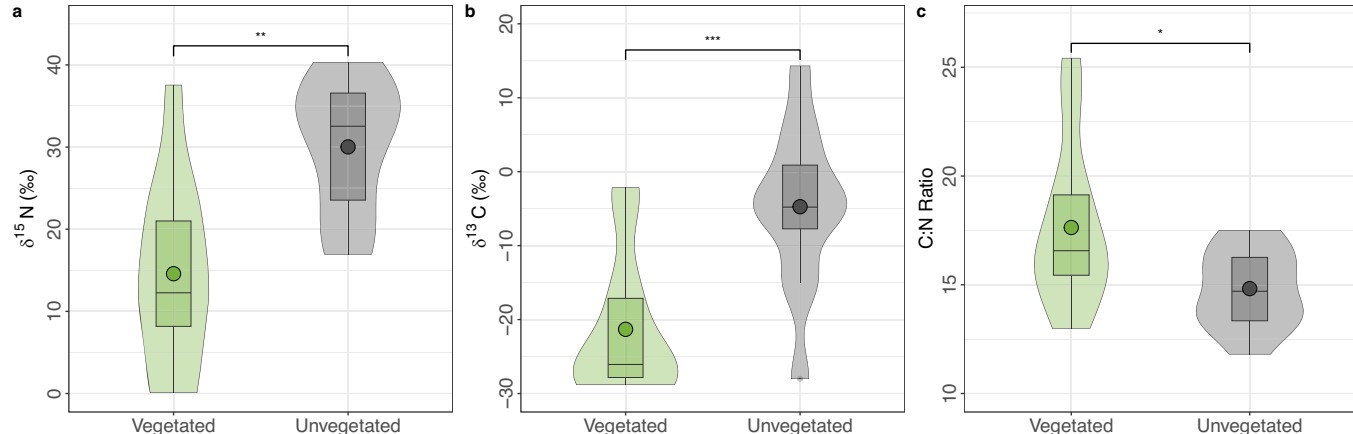

**Fig. 4 | δ $^{15}$N, δ $^{13}$C and C:N ratio.** Comparison of (**a**) δ 15 N (‰), (**b**) δ 13 C (‰), and (**c**) C:N ratio from soil samples with unvegetated and vegetated surface. Circles represent means and violin plots indicate the distribution of data. Asterisks indicate significance level: *$p < 0.01$; **$p < 0.005$; ***$p < 0.001$ (Two sample unpaired t-test).

linear regression to identify the change in concentration in the chamber headspace, including collar volume, over time and quality controlled by visual inspection. Fluxes were measured at 30 locations – 10 vegetated replicates and 20 unvegetated soil replicates – whenever weather permitted during July 2021 for a total of 263 measurements (8–10 measurements per collar). Measurements took place between 9:00 and 16:00 and the order was changed every day to ensure adequate temperature variation. The thermokarst polygon estimated coverage of 1.53% was based on UAV imagery of the study site (~5000 $m^2$; Supplementary Fig. 3). The UAV imagery was collected using a DJIP4 Multispectral drone. Surveys were flown at 12 m above-ground level resulting in a sampling resolution of 0.9 cm/pixel. On-the-ground accuracy was maintained to less than 1 cm, using a connected Realtime kinetic base station[81]. Post-processing relied on Pix4Dmapper. Unvegetated regions were digitized in QGIS software (Open Source Geospatial Foundation) using measurement locations as reference data. The estimated coverage was calculated as the ratio digitized area of unvegetated features to the area of the UAV imagery extent.

Ancillary measurements included soil surface temperature, bulk density, thaw depth, soil water content, stable isotope ratios, and C:N ratios. Soil water content, soil surface temperature, and thaw depth were measured at the time of each chamber measurement at the flux collar throughout the study period ($n = 263$). Soil measurements and samples were taken from the top 15 cm of the soil column, separated into three 5 cm layers. Soil water content was measured with a 300 TDR soil moisture meter (Fieldscout, USA) over the top 15 cm from the soil surface. The conditions during summer 2021 were within the ranges reported by the long-term mean[82], further supporting the representativeness of these measurements for emission rates. Soil surface temperature was measured with an TP7 infrared thermometer (Trotec, Germany). Thaw depth was measured with a small diameter metal rod inserted into the soil column until encountering resistance from the permafrost table. Bulk density was measured from soil samples from the top 15 cm of the soil column, collected at each collar location at the end of the study period, for a total of 30 samples. Soil samples were dried for 24 h at 60 °C in a drying oven and results expressed as dry weight per unit volume. WFPS was calculated by integrating bulk density, which represented the overall soil mass, and soil water content, indicating the water proportion according to ref. 83.

### Stable isotope analysis
Soil samples from the top 15 cm of the soil column were removed at both vegetated and unvegetated areas near where fluxes were measured using a handheld soil sampling corer (7 cm diameter, 15 cm height) at the end of the experiment. Soil samples consisted of four profiles with three depths (0–5 cm, 5–10 cm, & 10–15 cm) for a total of 24 samples, which were sieved for root removal. Samples were frozen and shipped to San Diego State University for stable isotope analysis. Samples were then separated into 5 cm depth segments (to check relationship with depth) using a band saw, placed in a drying oven at 65 °C for 48 h, then homogenized with a vibratory ball mill. Carbonate was not removed prior to analysis however, we would not expect this to contribute significantly to total C in these acidic, organic rich soils[67,84]. The abundance of $^{15}N$, $^{13}C$, and C:N ratios were measured using a continuous flow isotope ratio mass spectrometer (IRMS, Delta V Advantage, Thermo Fisher Scientific). A laboratory standard (USGS41, L-glutamic acid) was used as a reference material for the calibration of stable C and N measurements. Isotope values are reported in standard δ notation (‰) relative to Vienna PeeDee Belemnite ($\delta^{13}C$) and air-$N_2$ ($\delta^{15}N$).

### Statistics and data analysis
All data analyses were performed in R software, version R 4.1.0[85]. Data organization was performed using the 'data.table' R package[86]. Repeated measures ANOVAs were used for site differences (unvegetated, vegetated) for both $N_2O$ and $CO_2$ fluxes using collar location as a random variable to represent hierarchical structure, controlling for the pseudo replication

related to measuring the same plots multiple times during the summer. A series of linear mixed effects models were used to predict the variability in $FN_2O$. Model predictor variables included WFPS, soil temperature, thaw depth and various variable interactive terms. Linear mixed-effects model comparison showed that the models including soil temperature and WFPS as predictors for $FN_2O$ at unvegetated areas, along with an interactive term capturing their combined effect demonstrated superior predictive performance, as indicated by lower AIC relative to alternative models (Supplementary Table 1). $N_2O$ fluxes were square root transformed to meet the normality and homoscedasticity assumptions required for analyses. Assumptions of normality and homoscedasticity were verified with residual diagnostic tests. All model variables for were checked for multi-collinearity (VIF < 2.3; Tolerance statistic >0.4) using the 'olsrr' R package[87]. Graphics were generated using the 'ggplot2'[88], 'ggsignif'[89], 'cowplot'[90], and 'interactions'[91] packages. Two sample unpaired t-tests were used for comparisons of stable isotope content and C:N Ratios (Fig. 4).

### Reporting summary
Further information on research design is available in the Nature Portfolio Reporting Summary linked to this article.

## Data availability
All data that support the findings of this study are openly available at: https://doi.org/10.5281/zenodo.8391857.

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

## Acknowledgements
This work was funded by the NASA ABoVE Program (No. NNX16AF94A), the National Science Foundation—Office of Polar Programs (Nos. 1204263 and 1702797), NOAA CESSRST EPP (No. NA16SEC4810008), the European Union's Horizon 2020 research and innovation program (No. 629727890), the Natural Environment Research Council (NERC) UAMS Grant (No. NE/P002552/1) and European Research Council (ERC) Horizon 2020 Starting Grant FluxWIN, (No. 851181). J. Hashemi was partially supported by the Joint Doctoral Program in Ecology at San Diego State University and the University of California - Davis. We would like to thank the Ukpeaġvik Inupiat Corporation for access to conduct this research on UIC owned land. Additionally, we would like to extend our gratitude to reviewers for their time and insightful feedback, which greatly contributed to enhancing the quality of this study.

## Author contributions
J.H., W.C.O., and D.Z. conceived the work. J.H. designed and carried out field measurement campaign. J.H. carried out data analysis. J.H., D.A.L., K.A.A., S.J.D., A.K., K.L., L.V.D., W.C.O, and D.Z. contributed to data interpretation. J.H. wrote the manuscript with input from D.A.L., K.A.A., S.J.D., A.K., K.L., L.V.D., W.C.O, and D.Z. Visualization was done by J.H. Funding acquisition by W.C.O. and D.Z.

## Funding

## Competing interests

The authors declare no competing interests.
