## [Transparent Peer Review file · Communications Earth & Environment]

Thermokarst landscape exhibits large nitrous oxide emissions in Alaska's coastal polygonal tundra

Corresponding Author: Dr Josh Hashemi

Version 0:

Decision Letter:

Dear Dr Hashemi,

Your manuscript titled "Thermokarst landscape features exhibit large nitrous oxide emissions in an Arctic coastal polygonal tundra" has now been seen by 2 reviewers, and we include their comments at the end of this message. They find your work of interest, but some important points are raised. We are interested in the possibility of publishing your study in Communications Earth & Environment, but would like to consider your responses to these concerns and assess a revised manuscript before we make a final decision on publication.

We therefore invite you to revise and resubmit your manuscript, along with a point-by-point response that takes into account the points raised. In particular, we ask that you tone down your claims to reflect the uncertainties and limitations in your analysis. Please highlight all changes in the manuscript text file.

Please use the following link to submit your revised manuscript, point-by-point response to the referees' comments (which should be in a separate document to any cover letter), a tracked-changes version of the manuscript (as a PDF file) and the completed checklist:

Link Redacted

We hope to receive your revised paper within six weeks; please let us know if you aren't able to submit it within this time so that we can discuss how best to proceed. If we don't hear from you, and the revision process takes significantly longer, we may close your file. In this event, we will still be happy to reconsider your paper at a later date, as long as nothing similar has been accepted for publication at Communications Earth & Environment or published elsewhere in the meantime.

Please do not hesitate to contact us if you have any questions or would like to discuss these revisions further. We look forward to seeing the revised manuscript and thank you for the opportunity to review your work.

Best regards,

Clare Davis, PhD
Senior Editor
Communications Earth & Environment

www.nature.com/commsenv/
@CommsEarth

EDITORIAL POLICIES AND FORMATTING

Editorial Policy: [Policy requirements](https://www.nature.com/documents/nr-editorial-policy-checklist.pdf) (Download the link to your computer as a PDF.)

Furthermore, please align your manuscript with our format requirements, which are summarized on the following checklist: [Communications Earth & Environment formatting checklist](https://www.nature.com/documents/commsj-phys-style-formatting-checklist-article.pdf)

and also in our style and formatting guide [Communications Earth & Environment formatting guide](https://www.nature.com/documents/commsj-phys-style-formatting-guide-accept.pdf) .

*** DATA: Communications Earth & Environment endorses the principles of the Enabling FAIR data project (<http://www.copdess.org/enabling-fair-data-project/>). We ask authors to make the data that support their conclusions available in permanent, publically accessible data repositories. (Please contact the editor if you are unable to make your data available).

All Communications Earth & Environment manuscripts must include a section titled "Data Availability" at the end of the Methods section or main text (if no Methods). More information on this policy, is available at <http://www.nature.com/authors/policies/data/data-availability-statements-data-citations.pdf>.

If a community resource is unavailable, data can be submitted to generalist repositories such as [figshare](https://figshare.com/) or [Dryad Digital Repository](http://datadryad.org/). Please provide a unique identifier for the data (for example a DOI or a permanent URL) in the data availability statement, if possible. If the repository does not provide identifiers, we encourage authors to supply the search terms that will return the data. For data that have been obtained from publically available sources, please provide a URL and the specific data product name in the data availability statement. Data with a DOI should be further cited in the methods reference section.

REVIEWER COMMENTS:

Reviewer #1 (Remarks to the Author):

The manuscript by Hashemi et al. presents data on N₂O and CO₂ fluxes as well as auxiliary variables (such as soil water content, bulk soil C and N content and isotopic signatures) from vegetated and unvegetated locations in a polygonal tundra landscape at the North Slope of Alaska. Specifically, the authors find that unvegetated polygon centers emit N₂O at high rates, thereby identifying a new N₂O source in the Arctic. I have reviewed a previous version of this manuscript, and the manuscript has much improved during the revision. The authors have addressed many of my previous comments. I am satisfied with most of the revisions but have a few points remaining where I would like to ask for clarification, as well as some additional concerns that I recommend are addressed before publication.

- 1) Explanation and background knowledge on the dominant N₂O-producing processes and interactions with soil redox conditions have been added as suggested, much improving the text. In some cases, the discussion remains vague when it comes to the underlying processes (see line edits below for some examples), and I recommend to be a bit more specific to provide all necessary background information to the reader and to improve logical argumentation.
- 2) I still think the single comparison to wetlands in the abstract is misleading. I would suggest to additionally compare the measured flux rates to other Arctic N₂O hotspots that are more similar to the ones reported here (unvegetated, well-drained). Similarly, comparison to other unvegetated hotspots in tundra would be useful to add in the text (L129) to place results into context (e.g. Elberling et al 2010, Repo et al 2009, Marushchak et al 2011). The authors use the synthesized data from unvegetated soils in the permafrost region for comparison, but these also include data from low emission sites in rocky, high-Arctic terrain or mineral soils with low C/N ratio, not comparable to this study.
- 3) This paper does identify a novel N₂O source in the Arctic, undermined with detailed data on site and soil environmental

variables, which is a very valuable contribution to improving our understanding of N₂O dynamics from arctic ecosystems, contributing to upscaling and modeling efforts. At the same time, the paper also reports CO₂ fluxes, to put the measured N₂O flux rates in context of this GHG, which is important. But drawbacks are that process-based data are lacking, and that the larger implication of these findings remain elusive, also due to lack of available high-resolution data products to map the area contribution of the here studied thermokarst features, as the authors point out. Also, the authors argue that the high N₂O emissions from unvegetated polygon soils are driven by lack of plant-microbe competition for available inorganic N resources. While this makes sense and was confirmed by other studies, this point seems overemphasized given that scientific evidence is limited to C/N ratios and bulk soil d¹⁵N values. Since no inorganic N-pools, N-cycling process rates or microbial functioning were investigated, I would recommend to briefly acknowledge other potential drivers behind the differences in flux rates that can be based on the data, if possible. This would help to explore the novelty of this paper a bit further and set it apart from previously published literature. For example, in the context of C:N ratios (L205), could C-limitation also be an issue for microbial N-cycling patterns observed here? Could the differences between d¹⁵N values collected at different depths be explored a bit more, also in the context of the contribution of nitrification and denitrification to the overall flux? Again, I think the key interpretation of findings is correct, but the paper would benefit from a more balanced discussion, and from clearly stating the drawbacks as well (e.g., what data and data products are needed to improve understanding of such N₂O sources at the landscape scale? Are measurements of non-thermokarst features and wetter areas needed for landscape scale estimates?). Mentioning these shortcomings does not decrease the value of the study in my opinion, but provides a valuable resource for designing future studies. Most of this information is already there and would just need a few additions and a bit of restructuring.

4) What about the effect of the interaction between WFPS (or water content) and temperature or other variables; were interactive effects tested? It seems mixed-effects models were used, but Table S1 only reports the effect of individual predictors. Could low surface WFPS co-occur with high temperatures -> promoting denitrification in warmer, deeper layers, and that way explain why the highest fluxes were observed with low WFPS as opposed to previous studies (argument against nitrification as main driver of observed flux rates)? In that context, a time series of WFPS and temperature as (supplementary) figure may be useful.

5) Even after the revision it is not quite clear to me how CO₂ fluxes are reported. If I understand correctly, the reported means are derived from the flux measurements during daytime, and no interpolation of CO₂ fluxes based on temperature and light relationships was done. In that case, I suggest caution when stating the ecosystem is a sink or source (e.g. line 145 onwards), since the nighttime source, when plant photosynthesis is absent, was not considered. Please state clearly that only flux rates, measured during daytime, are considered, which do not amount to daily cumulative flux rates. Due to this reason, I also think that the comparison of the GWPs of CO₂ vs. N₂O (same paragraph) needs to be rephrased. The relative effect of CO₂ fluxes is probably larger if nighttime CO₂ emissions were considered. An alternative is to add interpolation functions for CO₂ fluxes to the paper and keep the current wording.

6) Reported values for auxiliary data: the d¹⁵N and d¹³C values have a wide range (and high positive values for d¹⁵N), can you please double-check those? Was carbonate removed before analysis? This can affect d¹³C values. pH can be an indicator for carbonate, and is also important for interpretation of N₂O fluxes. For a range of typical bulk soil d¹⁵N values please see for example Amundson et al. Along the same lines, please check the values for bulk density (L161): these seem too high to be in the unit g/cm³ and do not correspond to the figures. Also, the reported values for C and N (L270) are unrealistic, this would mean a C/N ratio of 0. Are these numbers simply swapped?
Amundson, R., Austin, A.T., Schuur, E.A., Yoo, K., Matzek, V., Kendall, C., Uebersax, A., Brenner, D. and Baisden, W.T., 2003. Global patterns of the isotopic composition of soil and plant nitrogen. *Global biogeochemical cycles*, 17(1).

7) Figures: please reconsider the choice of putting a (linear) regression line to some of the scatterplot figures which do not have a trend.

Line edits

L29-34: please reword in light of my comments above.

L70: but only if NO₃⁻ is present

L75: this is the upper estimate. It would be more conservative (and realistic) to provide the range.

L77/78: Some rephrasing needed. Earlier you mention that the main factor regulating N₂O production and release is temperature, limiting N-mineralization rates and thus substrate supply. This is the case not only during the period soils are frozen, but also during the relatively cool arctic growing seasons.

L89: and a percent estimate for the entire Arctic tundra/northern circumpolar PF region?

L105: the mentioned peat circles are not located in Siberia, but the Eastern European part of Russia / Western Russian Arctic. Please correct.

L108-111: seems like discussion?

L122: "support differences in biogeochemistry" is vague, please be more specific.

L123: higher instead of stronger?

L131-133: comparison to other (unvegetated) N₂O hotspots in the Arctic would fit well in this context.

L140: please be more specific as to how N from deeper layers would promote N₂O production. Enhanced mineralization and mineral N availability from nitrification and denitrification?

L141: If the dominant vegetation on the polygon rims were mosses and lichen, does that mean that vascular plants were mostly absent? What drives the assumed difference in inorganic N availability if there is no root uptake, can this be attributed mainly to BNF?

L158: bulk density and water content are not really "components" of WFPS, please rephrase.

L168: please be more specific: dominant pathway for what? N₂O production?

L176: don't forget NO₃- availability as the prerequisite for N₂O production during denitrification. See for example Marushchak et al.

Marushchak, M.E. et al, 2021. Thawing Yedoma permafrost is a neglected nitrous oxide source. Nature communications, 12(1), p.7107.

L188: This statement is unclear, can you please be more specific? Indicated larger losses of gaseous N species through microbial processes such as nitrification and denitrification?

L191-194: repetitive

L229-231: Would it be possible to use the same flux rates, expressed either per day or per hour for easy comparability?

Fig. 3: since WFPS is a function of bulk density, panel B seems redundant.

Reviewer #2 (Remarks to the Author):

This is an interesting paper reporting on high N₂O emissions from barren wetland soils in the Arctic, adding new data to some previous observations (see e.g. Voigt et al. Nature Reviews). However, the study also has some strong limitations, which are not yet sufficiently outlined to the reader, e.g. lack of mineral N or diurnal N₂O flux measurements (can observed diurnal values be extrapolated to daily or seasonal fluxes, as previous studies with high N₂O emissions usually also show high diurnal variations). In particular, in Arctic environments and with barren soils one would expect significant diurnal variations (that the bivariate plots of temperature versus N₂O flux do not show this is not surprising, as from day to day and plot to plot soil moisture also varies). Also, the time period with measurements is limited to the summer season and does not include the shoulder or non growing seasons. Due to the lack of supporting data on microbial processes or mineral N dynamics, much of the manuscript is speculative. While the argumentation is mostly logical, it remains that supporting data are lacking to make conclusive statements. While I agree that some permafrost soils are likely hotspots for N₂O emissions, the paper would be significantly stronger if the authors were able to indicate what percentage of the landscape, specifically the one where the measurements were made, could act as a hotspot for N₂O.

Line 33 Can you give the approximate contribution of this flux source for the landscape where you made your measurements (only for the summer season, and this should also be mentioned in the abstract)?

Line 123 Here "n" refers to number of plots.

Line 145 The violine plot for NEE (Fig. 2a) indicates that sometimes net CO₂ uptake was observed for unvegetated sites. How do you explain this?

Line 155 This is a statement that is not based on data, as information on soil inorganic N concentrations is lacking, and this should be clearly stated.

Line 159 Given the magnitude of fluxes from unvegetated soils, one would expect significant diurnal variations in fluxes. Have diurnal fluxes been measured? And what is the magnitude of the nighttime fluxes? Can you provide a graph showing the magnitude of fluxes as a function of time of measurement?

Line 161-162 Something is wrong with the bulk density values. Typical values for peat soils are 0.07-0.37 g cm⁻³ for the top 10 cm and 0.07-1.18 g cm⁻³ for 40-50cm depth. Please check your data

Line 168 While I agree that denitrification is likely the dominant N₂O production process, you contradict your statement in line 48 where you state that nitrification is the dominant process of N₂O production in Arctic soils. Both remain speculation, as there is no supporting data to discriminate between source processes (e.g. N₂O isotopomer measurements).

Line 188 An enrichment of the ¹⁵N signal of the bulk soil can be associated with nitrification (if nitrate is leached), with denitrification (if high losses of N₂O and N₂ produced by denitrification occur) or with NH₃ volatilization. Since nitrate leaching from the soil surface to deeper soil layers cannot be excluded (and can only be determined by measurements and ¹⁵N profile analysis), this remains a rather vague statement. It might be possible to discuss some more possibilities why unvegetated patches show such a high ¹⁵N enrichment (I assume that besides denitrification also nitrate leaching into deeper soil layers and NH₃ volatilization might play a significant role). This also leads to the question where in the soil the N₂O is produced.

Line 207 these values refer to total N, i.e. dominated by organic N

Line 215 -227 Would it be possible to provide an estimate for your study site and a few hectares? This would be very meaningful and provide some more substance.

Line 232-234 These are snapshot measurements, valid only for a specific day, and one must be careful to imply that they can be scaled to entire landscapes and entire seasons. While it is nice to mention this, it only emphasizes that more measurements are needed to get better estimates (by the way, working with two-digit numbers in such a context is probably not appropriate).

Line 244 I would recommend adding a paragraph about the limitations of the study toward the end of the manuscript, especially since many statements about source processes and regional significance are speculative.

Line 258 I don't understand the logic of why global warming must lead to increased emissions from barren soils. Will the area of barren soil increase with global warming? Or will the area of barren land shrink as vegetation growth increases? How much is the area of vegetated versus unvegetated land likely to change? Can you add to this information or give some hints?

Line 287 Please also provide information on the detection limit of N₂O and CO₂ fluxes and on calibration schemes and cross-sensitivities (e.g. between CO₂ and N₂O or water vapor)

Communications Earth & Environment is committed to improving transparency in authorship. As part of our efforts in this direction, we are now requesting that all authors identified as 'corresponding author' create and link their Open Researcher and Contributor Identifier (ORCID) with their account on the Manuscript Tracking System prior to acceptance. ORCID helps the scientific community achieve unambiguous attribution of all scholarly contributions. You can create and link your ORCID from the home page of the Manuscript Tracking System by clicking on 'Modify my Springer Nature account' and following the instructions in the link below. Please also inform all co-authors that they can add their ORCID to their accounts and that they must do so prior to acceptance.

Author Rebuttal letter: The author's response to these comments can be found at the end of this file.

Version 1:

Decision Letter:

Dear Dr Hashemi,

Please accept my sincere apologies for the delay in reaching a decision on your manuscript. Your manuscript titled "Thermokarst landscape features exhibit large nitrous oxide emissions in an Arctic coastal polygonal tundra" has now been seen by our reviewers, whose comments appear below. In light of their advice we are delighted to say that we are happy, in principle, to publish a suitably revised version in Communications Earth & Environment under the open access CC BY license (Creative Commons Attribution v4.0 International License).

We therefore invite you to revise your paper one last time to address the remaining concerns of our reviewers. At the same time we ask that you edit your manuscript to comply with our format requirements and to maximise the accessibility and therefore the impact of your work.

EDITORIAL REQUESTS:

****Please take care to match our formatting and policy requirements. We will check revised manuscript and return manuscripts that do not comply. Such requests will lead to delays. ****

SUBMISSION INFORMATION:

OPEN ACCESS:

Communications Earth & Environment is a fully open access journal. Articles are made freely accessible on publication under a [CC BY license](http://creativecommons.org/licenses/by/4.0) (Creative Commons Attribution 4.0 International License). This license allows maximum dissemination and re-use of open access materials and is preferred by many research funding bodies.

For further information about article processing charges, open access funding, and advice and support from Nature

Research, please visit <https://www.nature.com/commsenv/article-processing-charges>

At acceptance, you will be provided with instructions for completing this CC BY license on behalf of all authors. This grants us the necessary permissions to publish your paper. Additionally, you will be asked to declare that all required third party permissions have been obtained, and to provide billing information in order to pay the article-processing charge (APC).

Link Redacted

Best regards,

Alice Drinkwater, PhD
Associate Editor
Communications Earth & Environment

REVIEWERS' COMMENTS:

Reviewer #2 (Remarks to the Author):

The authors have done a very good job in revising their manuscript. They have addressed all my comments, in particular they now also provide an estimate of the area of barren soils in the studied landscape. I would strongly recommend including the graph of diurnal variations of N₂O fluxes on barren versus vegetated soils, at least in the Supplementary Information section. This graph clearly shows that diurnal variations in fluxes are significant (as is also evident from the strong temperature dependence). The diurnal dependence of N₂O fluxes (and the likely importance of diurnal variations in N₂O fluxes from barren soils) should, in my opinion, be explicitly mentioned around line 139 as it has significant implications for upscaling the magnitude of fluxes and should be pointed out as a knowledge gap in the abstract.

Reviewer #3 (Remarks to the Author):

The manuscript by Hashemi et al. presents interesting data on N₂O fluxes from vegetated and unvegetated locations in a polygonal tundra landscape at the North Slope of Alaska. A key finding is that unvegetated polygon centers emit N₂O at particularly high rates suggesting a hitherto ignored N₂O source in the Arctic. This is the first time I read this manuscript leaving the impression of a carefully revised and well written and structured manuscript. Only in a few places I suggest some further revisions to finish the work. That is, mostly minor typos that need correction, but one observation needs further attention. The finding that soil C in unvegetated polygons has a $\delta^{13}\text{C}$ value as high as -5 ‰ is very surprising and, to my knowledge, unrealistic if the only C is of plant origin. Do the authors have any other documentation supporting this unique finding? I acknowledge that acidic soils, as pointed out for this location, should hold no carbonates. But has this been checked? And what is the pH of the current soil? This should be reported if available. It's interesting that in vegetated plots $\delta^{13}\text{C}$ is much lower, below -21 ‰ but not as low as would be expected from purely C₃-plant carbon. Perhaps a mixing of the inorganic -5 ‰ C with planted derived C? Pure speculation, but I believed this needs some further attention as this also questions the validity of reporting C:N ratios under the assumption that everything is purely organic C (and N).

A few, minor typos:

Abstract, line 33: Give unit for emission (N₂O-N?)

Results and Discussion, line 130 and throughout: check the missing superscript on units.

Line 213 and throughout: The references to supplementary figures are not aligned correctly in mere places.

Line 262: Unit of flux

Line 388: Correct "Regression Assumptions"

Supplementary figure S2. Give unit of the Soil Water Content (volumetric?)

Author Rebuttal letter:

Referee Response Letter
Dear Reviewers,

We would like to thank you for the valuable time and effort you have invested in reviewing our manuscript. Your insightful comments and suggestions have significantly enhanced the quality of our work. We deeply appreciate both your commitment to the peer review process and your constructive feedback.

Reviewer #2 (Remarks to the Author):

The authors have done a very good job in revising their manuscript. They have addressed all my comments, in particular they now also provide an estimate of the area of barren soils in the studied landscape. I would strongly recommend including the graph of diurnal variations of N₂O fluxes on barren versus vegetated soils, at least in the Supplementary Information section. This graph clearly shows that diurnal variations in fluxes are significant (as is also evident from the strong temperature dependence). The diurnal dependence of N₂O fluxes (and the likely importance of diurnal variations in N₂O fluxes from barren soils) should, in my opinion, be explicitly mentioned around line 139 as it has significant implications for upscaling the magnitude of fluxes and should be pointed out as a knowledge gap in the abstract. Thank you for your feedback and time spent on this manuscript which has led to great improvement of the work.

The figure showing the fluxes as a function of time of day has been added to the supplementary section and alluded to within the body of the manuscript (L138-L141).

Reviewer #3 (Remarks to the Author):

The manuscript by Hashemi et al. presents interesting data on N₂O fluxes from vegetated and unvegetated locations in a polygonal tundra landscape at the North Slope of Alaska. A key finding is that unvegetated polygon centers emit N₂O at particularly high rates suggesting a hitherto ignored N₂O source in the Arctic. This is the first time I read this manuscript leaving the impression of a carefully revised and well written and structured manuscript. Only in a few places I suggest some further revisions to finish the work. That is, mostly minor typos that need correction, but one observation needs further attention. The finding that soil C in unvegetated polygons has a $\delta^{13}\text{C}$ value as high as -5‰ is very surprising and, to my knowledge, unrealistic if the only C is of plant origin. Do the authors have any other documentation supporting this unique finding? I acknowledge that acidic soils, as pointed out for this location, should hold no carbonates. But has this been checked? And what is the pH of the current soil? This should be reported if available. It's interesting that in vegetated plots $\delta^{13}\text{C}$ is much lower, below -21‰ but not as low as would be expected from purely C₃-plant carbon. Perhaps a mixing of the inorganic -5‰ C with planted derived C? Pure speculation, but I believed this needs some further attention

1

as this also questions the validity of reporting C:N ratios under the assumption that everything is purely organic C (and N).

Thank you very much for your feedback and effort in reviewing our manuscript. We agree that the $\delta^{13}\text{C}$ values in unvegetated soils are notably high. Unfortunately, we do not have pH or carbonate content data for the soils in this study, but this is an interesting topic and we would like to look more into this in future campaigns.

Our interpretation of pH is based on pH values from the general area as reported in various studies. We recognize that it is possible that these soils may and be more alkaline than the general area have localized carbonate accumulation. However, we also expect that the organic $\delta^{13}\text{C}$ signature is influenced by microbial products derived from older labile plant compounds, combined with the preferential loss of lighter carbon over time and a lack of seasonal inputs, unlike the surrounding vegetated areas.

We also agree that vegetated areas exhibit slightly elevated $\delta^{13}\text{C}$ values, and that this could be due to mixing with nearby unvegetated soils caused by freeze-thaw cycles and the close proximity. Some support for this may be the median value being around -26‰ , consistent with other literature, with higher values found in deeper soil samples where heaving and mixing would occur at a greater intensity.

We have added language to discuss both of these scenarios, or a combination of the two, to the manuscript (L198-L201; L218-L221).

A few, minor typos:

Abstract, line 33: Give unit for emission (N₂O-N?)

Thank you. Revised. (L32)

Results and Discussion, line 130 and throughout: check the missing superscript on units.

Thank you. Revised. (L129-L134; L272)

Line 213 and throughout: The references to supplementary figures are not aligned correctly in mere places.

Thank you. Revised throughout the manuscript.

Line 262: Unit of flux

Thank you. Revised. (L272)

Line 388: Correct "Regression Assumptions"

Thank you. Revised. (L400)

Supplementary figure S2. Give unit of the Soil Water Content (volumetric?)

Thank you. Revised.

Reviewers' comments with responses

Dear Reviewers,

We would like to express our gratitude for your contribution as a reviewer for our manuscript titled "Thermokarst landscape features exhibit large nitrous oxide emissions in an Arctic coastal polygonal tundra". Your insights and suggestions have played a pivotal role in shaping the development of our work, and we are very appreciative of the time and effort you dedicated to evaluating our research.

We have considered each of your points and suggestions, and as a result, we have made revisions to improve the overall coherence and rigor of the findings. We believe that the manuscript has benefited greatly as a result of your contributions. We have responded to all comments and incorporated your feedback into the revised version of the manuscript.

Please find attached the point-by-point responses below. We welcome any additional feedback or suggestions you may have. Thank you once again for your time, expertise, and commitment to the peer review process.

REVIEWER COMMENTS:

Reviewer #1 (Remarks to the Author):

The manuscript by Hashemi et al. presents data on N₂O and CO₂ fluxes as well as auxiliary variables (such as soil water content, bulk soil C and N content and isotopic signatures) from vegetated and unvegetated locations in a polygonal tundra landscape at the North Slope of Alaska. Specifically, the authors find that unvegetated polygon centers emit N₂O at high rates, thereby identifying a new N₂O source in the Arctic. I have reviewed a previous version of this manuscript, and the manuscript has much improved during the revision. The authors have addressed many of my previous comments. I am satisfied with most of the revisions but have a few points remaining where I would like to ask for clarification, as well as some additional concerns that I recommend are addressed before publication.

Thank you for your feedback - we very much appreciate the time and effort you've invested on the improvement of this manuscript for both iterations. We have attempted to address each of the listed points below.

1) Explanation and background knowledge on the dominant N₂O-producing processes and interactions with soil redox conditions have been added as suggested, much improving the text. In some cases, the discussion remains vague when it comes to the underlying processes (see line edits below for some examples), and I recommend to be a bit more specific to provide all necessary background information to the reader and to improve logical argumentation.

Thank you for the comment and suggestions. We have added further, more specific process-related background in response to the line edit suggestions below.

2) I still think the single comparison to wetlands in the abstract is misleading. I would suggest to additionally compare the measured flux rates to other Arctic N₂O hotspots that are more similar to the ones reported here (unvegetated, well-drained). Similarly, comparison to other unvegetated hotspots in tundra would be useful to add in the text (L129) to place results into context (e.g. Elberling et al 2010, Repo et al 2009, Marushchak et al 2011). The authors use the synthesized data from unvegetated soils in the permafrost region for comparison, but these also include data from low emission sites in rocky, high-Arctic terrain or mineral soils with low C/N ratio, not comparable to this study.

We have added to the abstract to reflect that these fluxes are of a similar magnitude to those identified from barren areas in peat plateaus and palsa mires. This information has also been added in more detail in the text with specific magnitudes (L28).

3) This paper does identify a novel N₂O source in the Arctic, undermined with detailed data on site and soil environmental variables, which is a very valuable contribution to improving our understanding of N₂O dynamics from arctic ecosystems, contributing to upscaling and modeling efforts. At the same time, the paper also reports CO₂ fluxes, to put the measured N₂O flux rates in context of this GHG, which is important. But drawbacks are that process-based data are lacking, and that the larger implication of these findings remain elusive, also due to lack of available high-resolution data products to map the area contribution of the here studied thermokarst features, as the authors point out. Also, the authors argue that the high N₂O emissions from unvegetated polygon soils are driven by lack of plant-microbe competition for available inorganic N resources.

While this makes sense and was confirmed by other studies, this point seems overemphasized given that scientific evidence is limited to C/N ratios and bulk soil δ¹⁵N values. Since no inorganic N-pools, N-cycling process rates or microbial functioning were investigated, I would recommend to briefly acknowledge other potential drivers behind the differences in flux rates that can be based on the data, if possible. This would help to explore the novelty of this paper a bit further and set it apart from previously published literature. For example, in the context of C:N ratios (L205), could C-limitation also be an issue for microbial N-cycling patterns observed here? Could the differences between δ¹⁵N values collected at different depths be explored a bit more, also in the context of the contribution of nitrification and denitrification to the overall flux? Again, I think the key interpretation of findings is correct, but the paper would benefit from a more balanced discussion, and from clearly stating the drawbacks as well (e.g., what data and data products are needed to improve understanding of such N₂O sources at the landscape scale? Are measurements of non-thermokarst features and wetter areas needed for landscape scale estimates?). Mentioning these shortcomings does not decrease the value of the study in my opinion, but provides a valuable resource for designing future studies. Most of this information is already there and would just need a few additions and a bit of restructuring.

Thank you for the feedback. We have tried to dial back language that indicates a certainty that site differences in emissions is due to a lack of plant competition for inorganic N. We have attempted to acknowledge other potential

drivers as suggested by the reviewer as well as explore the relationship of d15N with soil depth (L198; 212-222).

We have added a very limited estimate of the feature abundance at the study site derived from UAV imagery (~1.5% of land surface over a half hectare). This context has been inserted throughout the manuscript (L31; L259-263; L339-345) and a supplementary figure of the UAV imagery/feature abundance has been added (Fig. S3)

Also, language has been added discussing the limitations of the findings presented here with emphasis placed on what is needed to improve understanding of the implications of this study/future directions. (L271-294)

4) What about the effect of the interaction between WFPS (or water content) and temperature or other variables; were interactive effects tested? It seems mixed-effects models were used, but Table S1 only reports the effect of individual predictors. Could low surface WFPS co-occur with high temperatures -> promoting denitrification in warmer, deeper layers, and that way explain why the highest fluxes were observed with low WFPS as opposed to previous studies (argument against nitrification as main driver of observed flux rates)? In that context, a time series of WFPS and temperature as (supplementary) figure may be useful.

Yes, There is an interaction with WFPS/SWC & soil temperature. As you have indicated, high emissions occurring at low WFPS also tend toward warmer soil temperatures. We agree that this supports the occurrence of denitrification in deeper layers and have included this reasoning in the manuscript (LXX). This comment has made us reconsider the way we have approached using the mean SWC with bulk density to arrive at WFPS per collar. We have instead applied the rather static bulk density to unaveraged SWC to show the WFPS per unaveraged flux instead of mean flux per collar. We have updated the figures and language on this throughout the manuscript. We have also revised Table S1 to show only models without means and have included models with interaction terms. Thank you for this suggestion that has significantly improved the results. (L167-192; L378-386; Fig. 3)

5) Even after the revision it is not quite clear to me how CO₂ fluxes are reported. If I understand correctly, the reported means are derived from the flux measurements during daytime, and no interpolation of CO₂ fluxes based on temperature and light relationships was done. In that case, I suggest caution when stating the ecosystem is a sink or source (e.g. line 145 onwards), since the nighttime source, when plant photosynthesis is absent, was not considered. Please state clearly that only flux rates, measured during daytime, are considered, which do not amount to daily cumulative flux rates. Due to this reason, I also think that the comparison of the GWPs of CO₂ vs. N₂O (same paragraph) needs to be rephrased. The relative effect of CO₂ fluxes is probably larger if nighttime CO₂ emissions were considered. An alternative is to add interpolation functions for CO₂ fluxes to the paper and keep the current wording.

Thank you for your feedback. CO₂ and N₂O fluxes were measured concurrently, but only during “daytime” (9:00 to 16:00). Diurnal measurements for CO₂ were not conducted, nor were diurnal models applied. We feel that using interpolated CO₂ diurnals would not be appropriate as the comparison would

assume no diurnal variability for N₂O or require modeling N₂O diurnals as well, introducing more uncertainty. We do agree that clearly stating that the comparison does not account for diurnal variability is needed and this has been added (L171-174) as suggested by the reviewer.

6) Reported values for auxiliary data: the d15N and d13C values have a wide range (and high positive values for d15N), can you please double-check those? Was carbonate removed before analysis? This can affect d13C values. pH can be an indicator for carbonate, and is also important for interpretation of N₂O fluxes. For a range of typical bulk soil d15N values please see for example Amundson et al. Along the same lines, please check the values for bulk density (L161): these seem too high to be in the unit g/cm³ and do not correspond to the figures. Also, the reported values for C and N (L270) are unrealistic, this would mean a C/N ratio of 0. Are these numbers simply swapped?

Amundson, R., Austin, A.T., Schuur, E.A., Yoo, K., Matzek, V., Kendall, C., Uebersax, A., Brenner, D. and Baisden, W.T., 2003. Global patterns of the isotopic composition of soil and plant nitrogen. *Global biogeochemical cycles*, 17(1).

The bulk density decimal was in the wrong place and has now been corrected. Also, Yes - the C and N values were swapped - Now corrected. Thank you for catching these errors. (L187-188; L334)

We are confident the values for d15N and d13C are correct as the values for the standard (USGS40) are very close to the correct values. Carbonate was not removed prior to analysis but we would not expect this to contribute significantly to total C in these acidic, organic rich soils (Monhonval et al., 2023; Lipson et al., 2012). Language on these points has been added to the manuscript. (L402-404)

Monhonval et al., 2023., Mineral organic carbon interactions in dry versus wet tundra soils. *Geoderma*. 436, 116552, ISSN 0016-7061. <https://doi.org/10.1016/j.geoderma.2023.116552>.

Lipson, D. A., Zona, D., Raab, T. K., Bozzolo, F., Mauritz, M., and Oechel, W. C.: Water-table height and microtopography control biogeochemical cycling in an Arctic coastal tundra ecosystem, *Biogeosciences*, 9, 577–591, <https://doi.org/10.5194/bg-9-577-2012>, 2012.

7) Figures: please reconsider the choice of putting a (linear) regression line to some of the scatterplot figures which do not have a trend.

Thank you. Non-significant trend regression lines have been removed.

Line edits

L29-34: please reword in light of my comments above.

We have added that these are similar in magnitude to other Arctic unvegetated hotspots as recommended. (L28)

L70: but only if NO₃⁻ is present

Thank you. Revised. (L70)

L75: this is the upper estimate. It would be more conservative (and realistic) to provide the range.

Agreed. Revised. (L75)

L77/78: Some rephrasing needed. Earlier you mention that the main factor regulating N₂O production and release is temperature, limiting N-mineralization rates and thus substrate supply. This is the case not only during the period soils are frozen, but also during the relatively cool arctic growing seasons.

Thank you. Revised. (L76)

L89: and a percent estimate for the entire Arctic tundra/northern circumpolar PF region?

Thank you. Revised. (L89)

L105: the mentioned peat circles are not located in Siberia, but the Eastern European part of Russia / Western Russian Arctic. Please correct.

Thank you. Revised. (L104)

L108-111: seems like discussion?

Thanks, but we feel this briefly contextualizes the importance of the study early on and leads well to the results and discussion section.

L122: "support differences in biogeochemistry" is vague, please be more specific.

Thank you. Revised. (L120-122)

L123: higher instead of stronger?

Thank you. Revised. (L127)

L131-133: comparison to other (unvegetated) N₂O hotspots in the Arctic would fit well in this context.

Thank you. Revised. (L130-133)

L140: please be more specific as to how N from deeper layers would promote N₂O production. Enhanced mineralization and mineral N availability from nitrification and denitrification?

Thank you. Revised. (L145-149)

L141: If the dominant vegetation on the polygon rims were mosses and lichen, does that mean that vascular plants were mostly absent? What drives the assumed difference in inorganic N availability if there is no root uptake, can this be attributed mainly to BNF?

Vegetated areas are characterized by both moss-lichen and vascular plants. “Dominated” is misleading and has been rephrased. Thanks. (L149)

L158: bulk density and water content are not really “components” of WFPS, please rephrase.

Thank you. Removed.

L168: please be more specific: dominant pathway for what? N₂O production?

Thank you. Revised. (L178)

L176: don't forget NO₃⁻ availability as the prerequisite for N₂O production during denitrification. See for example Marushchak et al.

Marushchak, M.E. et al, 2021. Thawing Yedoma permafrost is a neglected nitrous oxide source. Nature communications, 12(1), p.7107.

Thank you. Revised. (L186)

L188: This statement is unclear, can you please be more specific? Indicated larger losses of gaseous N species through microbial processes such as nitrification and denitrification?

Thank you. Revised. (L196-197)

L191-194: repetitive

Thank you. Removed.

L229-231: Would it be possible to use the same flux rates, expressed either per day or per hour for easy comparability?

Thank you. Revised. (L249-250)

Fig. 3: since WFPS is a function of bulk density, panel B seems redundant.

Thank you. Panel B has been removed.

Reviewer #2 (Remarks to the Author):

This is an interesting paper reporting on high N₂O emissions from barren wetland soils in the Arctic, adding new data to some previous observations (see e.g. Voigt et al. Nature Reviews). However, the study also has some strong limitations, which are not yet sufficiently outlined to the reader, e.g. lack of mineral N or diurnal N₂O flux measurements (can observed diurnal values be extrapolated to daily or seasonal fluxes, as previous studies with high N₂O emissions usually also show high diurnal variations). In particular, in Arctic environments and with barren soils one would expect significant diurnal variations (that the bivariate plots of temperature versus N₂O flux do not show this is not surprising, as from day to day and plot to plot soil moisture also varies). Also, the time period with measurements is limited to the summer season and does not include the shoulder or non growing seasons. Due to the lack of supporting data on microbial processes or mineral N dynamics, much of the manuscript is speculative. While the argumentation is mostly logical, it remains that supporting data are lacking to make conclusive statements. While I agree that some permafrost soils are likely hotspots for N₂O emissions, the paper would be significantly stronger if the authors were able to indicate what percentage of the landscape, specifically the one where the measurements were made, could act as a hotspot for N₂O.

Line 33 Can you give the approximate contribution of this flux source for the landscape where you made your measurements (only for the summer season, and this should also be mentioned in the abstract)?

Yes, thanks. We have given an adjusted hectare estimate based on growing season median/mean emissions multiplied by the percent distribution as identified with UAV imagery (1.5%). We have included language on this in the abstract, results/discussion, and methods (L31; L259-263; L339-345) and a supplementary figure of the UAV imagery/feature abundance has been added (Fig. S3).

Line 123 Here “n” refers to number of plots.

Yes, this has been changed to “Number of measurement locations”. (L123-126)

Line 145 The violine plot for NEE (Fig. 2a) indicates that sometimes net CO₂ uptake was observed for unvegetated sites. How do you explain this?

Thanks. Very small uptake at the unvegetated site is observed with 2 measurements (1%) with low R² values associated with the slopes used in the flux calculations and we would interpret these as no flux.

Line 155 This is a statement that is not based on data, as information on soil inorganic N concentrations is lacking, and this should be clearly stated.

Thank you. Revised. (L180-183)

Line 159 Given the magnitude of fluxes from unvegetated soils, one would expect significant diurnal variations in fluxes. Have diurnal fluxes been measured? And what

is the magnitude of the nighttime fluxes? Can you provide a graph showing the magnitude of fluxes as a function of time of measurement?

Thank you for the comment. We did not make diurnal measurements as this study was conducted with manual rather than automated chamber measurements. As such, we do not have nighttime flux data. Below, please find the requested graph. Measurements were only taken between 9-16 and no apparent diurnal pattern is present for vegetated areas and though there is certainly an indication of a diurnal pattern, we do not have representation of the entire day. Language has been added on this topic (L171-174; L279; L283).

Line 161-162 Something is wrong with the bulk density values. Typical values for peat soils are 0.07-0.37 g cm⁻³ for the top 10 cm and 0.07-1.18 g cm⁻³ for 40-50cm depth. Please check your data

Thank you for catching this. The bulk density decimal was in the wrong place and has now been corrected (L187-188).

Line 168 While I agree that denitrification is likely the dominant N₂O production process, you contradict your statement in line 48 where you state that nitrification is the dominant process of N₂O production in Arctic soils. Both remain speculation, as there is no supporting data to discriminate between source processes (e.g. N₂O isotopomer measurements).

Thank you. We have revised in the introduction of the manuscript that both nitrification and denitrification are the dominant processes for N₂O production and subsequent emission in general (not specifically in Arctic regions) (L48).

Line 188 An enrichment of the ^{15}N signal of the bulk soil can be associated with nitrification (if nitrate is leached), with denitrification (if high losses of N_2O and N_2 produced by denitrification occur) or with NH_3 volatilization. Since nitrate leaching from the soil surface to deeper soil layers cannot be excluded (and can only be determined by measurements and ^{15}N profile analysis), this remains a rather vague statement. It might be possible to discuss some more possibilities why unvegetated patches show such a high ^{15}N enrichment (I assume that besides denitrification also nitrate leaching into deeper soil layers and NH_3 volatilization might play a significant role). This also leads to the question where in the soil the N_2O is produced.

Thank you for the feedback. We agree that the ^{15}N values are quite enriched and would interpret this as frequent losses from some combination of denitrification and nitrification. We do expect that nitrification could be partially responsible for emission patterns but would also expect some leaching to be limited due to the presence of permafrost. We would not expect ammonia volatilization to be significant due to the acidic soils characteristic of the region (Lipson et al., 2012).

For the question of where in the soil is N_2O produced, in previous work in this region, no clear association with denitrification genes were found with depth and are assumed to be rather evenly distributed throughout the soil column (Lipson et al., 2013). Gammaproteobacteria have been positively correlated with nitrate and found in more oxic areas (shallow soils, high topography). Denitrification is widespread in Gammaproteobacteria, providing indirect evidence that this are one of the groups responsible for denitrification, and occur where nitrate is abundant (Lipson et al., 2015)

We have added language on this topic to the manuscript (L198; 212-222).

Lipson, D. A., Zona, D., Raab, T. K., Bozzolo, F., Mauritz, M., and Oechel, W. C. 2012 Water-table height and microtopography control biogeochemical cycling in an Arctic coastal tundra ecosystem, *Biogeosciences*, 9, 577–591, <https://doi.org/10.5194/bg-9-577-2012>.

Lipson DA, Haggerty JM, Srinivas A, Raab TK, Sathe S, Dinsdale EA. 2013. Metagenomic insights into anaerobic metabolism along an Arctic peat soil profile. *PLoS One*. 8(5):e64659. doi: 10.1371/journal.pone.0064659. PMID: 23741360

Lipson DA, Raab TK, Parker M, Kelley ST, Brislawn CJ, Jansson J. 2015. Changes in microbial communities along redox gradients in polygonized Arctic wet tundra soils. *Environ Microbiol Rep*. 7(4):649-57. doi: 10.1111/1758-2229.12301.

Line 207 these values refer to total N, i.e. dominated by organic N

Thank you. Revised. (L248)

Line 215 -227 Would it be possible to provide an estimate for your study site and a few hectares? This would be very meaningful and provide some more substance.

We have obtained very limited (5000 m2) UAV imagery of the study region showing the distribution of the landscape features to be around 1.5%. This has been included as a supplementary figure (Fig. S3) and language has been added in the manuscript on this topic. Thank you. (L31; L259-263; L339-345)

Line 232-234 These are snapshot measurements, valid only for a specific day, and one must be careful to imply that they can be scaled to entire landscapes and entire seasons. While it is nice to mention this, it only emphasizes that more measurements are needed to get better estimates (by the way, working with two-digit numbers in such a context is probably not appropriate).

Thank you. Revised (We made adjustments to this number because we originally did not account for the difference in N₂O to N₂O-N)(L254).

Line 244 I would recommend adding a paragraph about the limitations of the study toward the end of the manuscript, especially since many statements about source processes and regional significance are speculative.

Agreed. We have added this to the manuscript. Thank you. (L271-294)

Line 258 I don't understand the logic of why global warming must lead to increased emissions from barren soils. Will the area of barren soil increase with global warming? Or will the area of barren land shrink as vegetation growth increases? How much is the area of vegetated versus unvegetated land likely to change? Can you add to this information or give some hints?

Thank you for the feedback. We have added a reference to a recent paper discussing thermokarst expansion associated with warming. We have made minor revisions in the paragraph starting at L239-243 to more clearly contextualize the logic that increased temperatures may lead to changes in active layer depth, hydrology, and permafrost table. All of which can have an effect of uneven ground subsidence in these regions often resulting in surface disturbance and exposure of barren soil. We are unaware of any literature an estimated change in distribution percentage of vegetated to barren soil for future climate scenarios. Though there will likely be vegetation growth in disturbed areas, we feel that the discussion of the land surface disturbance is needed.

Line 287 Please also provide information on the detection limit of N₂O and CO₂ fluxes and on calibration schemes and cross-sensitivities (e.g. between CO₂ and N₂O or water vapor)

Thank you. According to calibrations calculations done internally at Gaset, the LDLs are the same as the MCDCs already reported in the manuscript. All Gaset GT5000 Terra instruments are calibrated during manufacturing. There is not a typical routine calibration method/interval recommended to end users other than daily zeroing with N₂ gas. Gaset recommends a general maintenance inspection every 1 to 2 years, though the data presented here is from only months after the initial purchase of the GT5000 Terra from Gaset. Gaset maintains that there are no issues with cross-interference with gases, as the very nature of FTIR allows for unique regions with unique

peaks/characteristics within the measurement spectrum. We have added information on this topic to the manuscript (L320-326).